

# *Catostylus tagi* (Class: Scyphozoa, Order: Discomedusae, Suborder: Rhizostomida, Family: Catostylidae) life cycle and first insight into its ecology

Sonia K.M. Gueroun[1,2,3], Tatiana M. Torres[4], Antonina Dos Santos[5,6], Nuno Vasco-Rodrigues[7,8], João Canning-Clode[3,9] and Carlos Andrade[1,2,6]

[1] Mariculture Centre of Calheta, Calheta, Madeira, Portugal
[2] Madeira Oceanic Observatory-ARDITI/OOM, Funchal, Madeira, Portugal
[3] MARE–Marine and Environmental Sciences Centre, Agência Regional para o Desenvolvimento da Investigação Tecnologia e Inovação (ARDITI), Funchal, Madeira, Portugal
[4] Universität Bremen, Bremen, Germany
[5] Instituto Português do Mar e da Atmosfera (IPMA), Algés, Portugal
[6] CIIMAR-Interdisciplinary Centre of Marine and Environmental Research, Matosinhos, Portugal
[7] Oceanário de Lisboa, Lisbon, Portugal
[8] MARE–Marine and Environmental Sciences Centre, Instituto Politécnico de Leiria, Peniche, Portugal
[9] Smithsonian Environmental Research Center, Edgewater, USA

Corresponding author
Sonia K.M. Gueroun,
sgueroun@mare-centre.pt

## ABSTRACT

Jellyfish proliferations, which are conspicuous and natural events, cause blooms that may lead to severe consequences for anthropogenic activities and ecosystem structure and functioning. Although research during the last decade has focused on factors influencing the different jellyfish life stages, few species currently have their full life cycle known. In this context, we describe for the first time the developmental stages in the life cycle of *Catostylus tagi*, from planula to young medusa, reared in the laboratory. The species displays the typical Rhizostomida metagenetic life cycle. Mature scyphistomae display 16 tentacles and a total body length of 1.5 ± 0.2 mm. Only podocyst production and strobilation were observed. Strobilation, occurring continuously under laboratory conditions, was mainly polydisc. The eight-rayed typical ephyrae, with a total body diameter of 2.4 ± 0.4 mm at detachment, showed development typical of the Rhizostomida. As a first step in studying this species' ecology, we also present preliminary assessments of: (i) the influence of different temperature and salinity regimes on planulae survival, settlement and metamorphosis and (ii) the effect of temperature and diet on asexual reproduction. The results showed a high tolerance of planulae to a wide range of salinities (15‰ to 25‰), while polyp development was significantly faster at higher temperature (20–25 °C). Strobilation onset was 2–3 times faster at 20 °C (10.6 ± 5.4 to 15 ± 6.6 day at various tested diet) than at 15 °C (32.2 ± 3 day). Feeding was a key factor as unfed polyps never underwent strobilation during the trial. Finally, we present the spatial and seasonal distribution of *C. tagi* in the Tagus estuary (Portugal) in 2019, showing its occurrence throughout the year (except in April), with most observations recorded on the northern shoreline. As *C. tagi* shows the ability to form blooms and a

wide tolerance for temperature and salinity (for planulae and medusae stage), it is essential to understand its life cycle.

## INTRODUCTION

In recent decades, jellyfish have attracted much attention due to intense blooming events in coastal waters; such outbreaks often have negative repercussions on human activities (e.g., fisheries, aquaculture, tourism and power plants) and the structure and function of ecosystem (reviewed in *Purcell, Uye & Lo, 2007*; *Pitt, Welsh & Condon, 2009*). Nonetheless, no common consensus has been reached on whether jellyfish are actually increasing globally (*Condon et al., 2012*) or regarding the implications of various anthropogenic causes triggering these gelatinous organisms to proliferate (*Sanz-Martín et al., 2016*).

With some exceptions (*e.g.*, *Pelagia noctiluca*, *Rottini Sandrini & Avian, 1983*), Scyphozoan species are meroplanktonic with a bipartite life cycle. The pelagic medusa stage typically reproduces sexually, producing a free-swimming planula. After the planula attaches to a substrate, it grows into a sessile polyp. The polyp reproduces asexually through various budding modes (*e.g.*, lateral budding, budding from stolon, motile bud-like tissue particles), podocysts and *via* strobilation (*Arai, 1997*). Environmental factors (*e.g.*, temperature, prey supply) can affect both the sessile stage (*e.g.*, asexual reproduction timing and intensity) (*Yongze et al., 2016*) and the pelagic stage (*e.g.*, somatic growth and sexual maturation) (*Pitt & Kingsford, 2000*) of these species, making the study of both phases essential to scyphozoan biology. Understanding the influence of environmental factors on each life stage is crucial to better comprehending the dynamics of the species, while identifying early stages is primordial to detecting potential blooms. Paradoxically, the complete life cycle of scyphozoans has been described for less than 25% of known species (*Tronolone, Morandini & Migotto, 2002*). Generally, only the adult stage of scyphozoans is known (*Mills, 2001*; *Jarms & Morandini, 2019*).

The *Catostylus* genus consists of 10 valid species, occurring in temperate regions and subtropical and tropical regions (*Jarms & Morandini, 2019*). In the Atlantic, the genus is currently represented by three species: *C. cruciatus* (Lesson, 1830) in Brazil, *C. tagi* (Haeckel, 1869), whose distribution extends from the Atlantic coast of Europe (France, Spain and Portugal) to the West African coast (south of Congo) and *C. tripterus* (Haeckel, 1880), found off Equatorial Guinea. The distribution of *C. tagi* has been extended eastward since it was recorded as a non-indigenous species (NIS) in the Mediterranean Sea, in June 2010, in the Sicily Channel (*Nastasi, 2010*). Among the genus, only the life cycle of *C. mosaicus* has been investigated (*Pitt, 2000*).

*Catostylus tagi* is a common Scyphozoa in the Tagus estuary (Portugal), where juveniles and adults have been observed (GelAvista citizen science project). No records of ephyrae or polyps have been reported yet, and general information on this species' biology and

ecology remain scarce. To date, no studies the species' population dynamics, biology or ecology have ever been published. Only the medusa stage of *C. tagi* has been described, while its complete life cycle remains unknown. However, several studies on the biochemical properties of *C. tagi* have been conducted; these studies have found edible species for human consumption (*Amaral et al., 2018*; *Raposo et al., 2018*), as well as relevant collagen and antioxidant properties with potential as new bio-resources for the cosmetics and food sectors (*Calejo, Morais & Fernandes, 2009*; *Morais et al., 2009*). *Catostylus tagi* is moderately venomous (*GelAvista, 2021*) and is considered a harmless species in Portugal (*Morais et al., 2009*).

In the present study, we describe for the first time the complete life cycle of *C. tagi* based on fertilisation trials conducted in the laboratory. Additionally, we conducted a preliminary evaluation of the ecology of the species through assessments of the effect of: (i) temperature and salinity on the planula stage (pre-settlement survival, settlement and metamorphosis) and (ii) temperature and diet regimes on polyp asexual reproduction. Finally, we pioneer an overview of the spatial and seasonal distribution of *C. tagi* along the Tagus estuary showing the potential of the medusa stage to a broad range of temperature and salinity values.

## MATERIALS AND METHODS

### Ethics statement
The jellyfish *C. tagi* is not an endangered or protected species.

### Fertilisation
In early October 2019, six medusae were collected from the Tagus estuary (Portugal), near the Oceanário de Lisboa, and individually transferred to the laboratory within 30 mins in 15 L buckets. Specimens' sex and gonadal maturity were determined using a microscope. Of the six individuals, two were males while four were females. Gonads were extracted from the two most mature individuals, and gastric filaments and excess tissue removed. The female's and the male's bell diameters were 39.5 and 43 cm, respectively. The extracted gametes were mixed and incubated for 48 h with constant aeration in three L containers containing ultraviolet-treated artificial seawater (Red Sea's Premium salt®) at a salinity of 35‰ and room temperature (18 °C). After 48 h, the planulae were collected by filtering the medium on gradient mesh (200 and 55 μm). Plastic petri dishes, previously incubated for four days in natural seawater for biofilm development, were used as substrates. The petri dishes were placed at mid-height in 500 ml glass bowls, thus allowing the planulae to settle on both sides of the substrate.

### Culture maintenance
Planulae and polyps were incubated in artificial seawater (salinity 35‰) at 18 °C under a natural light/dark cycle. Collected ephyrae were maintained in several 500 ml jars. Once the metaephyra stage was reached, individuals were transferred to a 60 L pseudo-Kreisel under the same conditions (*i.e.*, temperature, salinity and feeding regime).

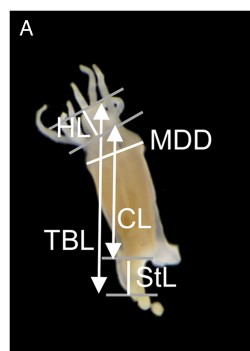
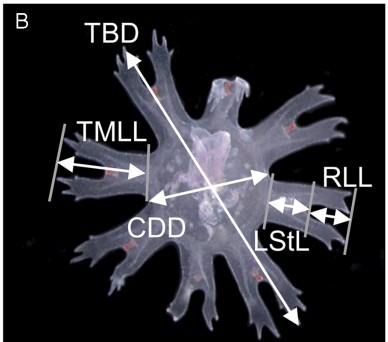

**Figure 1 Measuring points and measurements defined and taken in a polyp (A) and newly released ephyra (B).** TBL, total body length; CL, calyx length; HL, hypostome length; MDD, mouth disc diameter; StL, stalk length; TBD, total body diameter; CDD, central disc diameter; TMLL, total marginal lappet length, LStL, lappet stem length; RLL, rhopalial lappet length.

Polyps were fed daily with rotifers (*Brachionus plicatilis*) during the first week, after which newly hatched *Artemia* nauplii were added three times a week. Ephyrae and juveniles were fed rotifers (four times a day), *Artemia* (three times a day) and mashed mussel (once a day). Nauplii of AF *Artemia* Vietnam strain (small nauplii with high HUFA content, Inve Aquaculture NV®, Baasrode, Belgium) and enriched EG *Salt Lake Artemia fransiscana* (Inve Aquaculture NV®, Baasrode, Belgium) were used for ephyrae and juveniles, respectively. Every 2–3 days, a 50‰ water exchange was conducted.

## Anatomical analysis

Two different Stereomicroscopes (Leica® SAPO and Leica® M165C) were used to describe the various life stages, as well as to follow the development of the gastric system, manubrium, and marginal lappets of the newly released ephyrae (stage 0) through to the metaephyra stage (stage 7).

Measurements of the scyphistoma were taken following *Straehler-Pohl, Widmer & Morandini (2011)*: total body length (TBL), calyx length (CL), hypostome length (HL), mouth disc diameter (MDD) and stalk length (StL) (Fig. 1A). The following standard measurements were used for the young ephyrae (*Straehler-Pohl & Jarms, 2010*): total body diameter (TBD), central disc diameter (CDD), total marginal lappet length (TMLL), lappet stem length (LStL) and rhopalial lappet length (RLL) (Fig. 1B). Relative body dimensions (%) were calculated for scyphistomae (measurements compared with body length, *e.g.*, CL/TBL × 100, and calyx diameter, *e.g.*, MDD/CL × 100) and for ephyrae (measurements compared with body diameter, *e.g.*, CDD/TBD × 100, and lappet length, *e.g.*, RLL/TMLL × 100). A total of 11 scyphistomae and 20 ephyrae from five strobilae were measured.

## The effect of temperature and salinity on planula development

Two orthogonal treatment sets were established with three temperatures (15, 20 and 25 °C) and four different salinities (20, 25, 30 and 35‰) reflecting conditions recorded in the Tagus area (*Gameiro, Cartaxana & Brotas, 2007*; *Rodrigues et al., 2017*). Water was

prepared by diluting artificial seawater (35‰) with distilled water. Each treatment was tested with eighteen replicates (planulae) following the methods of *Conley & Uye (2015)* and *Takao & Uye (2018)*. A total of thirty-six polycarbonate culture plates, six-wells of ten ml, were prepared (three plates per experimental condition). Culture plates were filled with natural seawater four days prior to incubation in order to allow for biofilm development. Acclimatation of the planulae to lower salinities (30, 25, and 20‰) was done in a step-wise fashion, soaking the planulae in water of each decreasing salinity for 5 min until the target salinity was reached. Over six days, planulae were surveyed daily with a stereomicroscope (Zeiss® Stemi 305). Life stages of the planulae were recorded in the following manner: Dead; Stage 0: settled but no tentacles; Stage 1: 1–4 tentacles; Stage 2: 5–7 tentacles; Stage 3: 8–16 tentacles. None of the polyps were fed during this trial.

## The effect of temperature and feeding regimes on asexual reproduction

A total of two orthogonal treatment sets were established with two temperatures (15 and 20 °C) and three feeding regimes, comprising groups being fed rotifers (*Brachionus plicatilis*; $R_{group}$) or *Artemia* sp. nauplii ($A_{group}$) or being unfed ($U_{group}$). Eighteen polyps were tested using of each treatment combination. A total of one polyp was placed per well into 6-well polycarbonate culture plates (three plates per experimental condition) filled with 10 ml of the artificial seawater (35‰). A water bath was used to maintain the designed temperatures. The photoperiod was maintained at 12 h light:12 h dark. After allowing the polyps 1 week to reattach and acclimate to the experimental temperatures, newly hatched *Artemia* nauplii and *B. plicatilis* were fed in excess every 2 days. After a feeding period of 1.5 h, the wells were cleaned with swabs and uneaten food was discarded, while the seawater was replaced with new water of the same temperature. This feeding protocol provided saturating prey briefly, resulting in equal feeding in all treatments by minimising enhanced feeding at warmer temperatures (*Ma & Purcell, 2005*). Specimens in the unfed treatment never received food, although the water was exchanged similar to other treatment groups. Polyps were examined daily for strobilation and ephyrae release and twice a week for podocyst production. After enumeration, new ephyrae were removed but not the new podocysts. The experiment lasted 33 days.

Several response variables were defined for analysis, comprising: the number of podocysts produced by the polyps; the time from the beginning of the experiment to strobilation onset, *i.e.*, the "pre-strobilation" period (*pre-str*); the time from the beginning of strobilation to the release of the first ephyrae, *i.e.*, the "bet-strobilation" period (*bet-str*); the time from the first release of ephyrae to the release of the last ephyrae, *i.e.*, the "strobilation period" (*str*) the number of ephyrae produced for each strobilation event.

## Community science data

This study presents data on *C. tagi* sightings from Tagus estuary for the year 2019 gathered in the GelAvista Project's scope (gelavista.ipma.pt), mainly based on the GelAvista smartphone App. The project is a citizen science program that provides information on jellyfish' presence in Portugal through volunteer contributions of jellyfish sightings *via* the

GelAvista smartphone application, email address, and Facebook page. The collected data include GPS location, date, and hour of sighting and the approximate number of specimens spotted. Species identification is made through the examination of photographs or videos. A confidence level was assigned to all reports, taking into account the veracity and sufficiency of the information received.

## Statistical analysis

Since the measurements on the planulae consisted of repeated measures of binary outcomes, data on the planktonic stage were assigned to durations from the experiment onset to settlement time (maximum of 160 h) (*Takao & Uye, 2018*). The combined effect of temperature and salinity on the planktonic duration was tested by two-way ANOVA followed by Tukey pair-wise comparison. Data were square-root transformed to meet the residuals homogeneity assumption.

The *pre-str* data were analysed with generalised linear models for counts data. Due to excessive zero in the data, a zero-inflated model was used (*Zuur et al., 2009*). When zero-inflated Poisson model presented overdispersion, the model was corrected with the standard errors using a quasi-GLM model and a third modelwas fitted with a negative binomial distribution. The best models was selected based on the AIC and BIC values (*Zuur et al., 2009*).

Since only two combinations produced ephyrae, *bet-str*, *str* and ephyra released by strobilation event were analysed with a *T* or Wilcox ranking test depending on the variance homogeneity.

Data were analysed with the free R platform (version 3.0.2; R Development Core Team 2011) using *car* (*Fox & Weisberg, 2019*) and *glmmTMB* (*Brooks et al., 2017*).

# RESULTS

## Life cycle

*Catostylus tagi* displayed a typical metagenetic life cycle including scyphistoma and ephyra phases (Fig. 2). The first polyps were observed approximately 96 h after the planulae had been added to the culture plates. The young polyps were translucent-white, cone-shaped, and typically had four tentacles. Mature scyphistoma (Fig. 2A) had 16 tentacles in a single whorl around a slightly sunken mouth disc. The four-lipped hypostome was short, $325 \pm 64$ μm ($\approx 22\%$ TBL) and club-shaped. The calyx had an elongated cup shape. Scyphistoma colour varied from white to pale orange depending on the feeding. Measurements of *C. tagi* mature scyphistoma are summarized in Table 1.

The scyphistoma proliferated asexually *via* podocysts (Fig. 2A); this proliferation was observed starting as a periderm-enclosed podocyst. The podocysts were typically yellow or brown. A finger-shaped stolon developed from the lower part of the stalk and attached to the substrate allowing the scyphistoma to shift over. No other reproduction modalities, such as lateral budding by stolon, lateral scyphistoma budding or pedalocysts were observed.

Both monodisc (producing one ephyra) and polydisc (producing multiple ephyrae) strobilation were observed; monodisc strobilation was observed only once. At the first
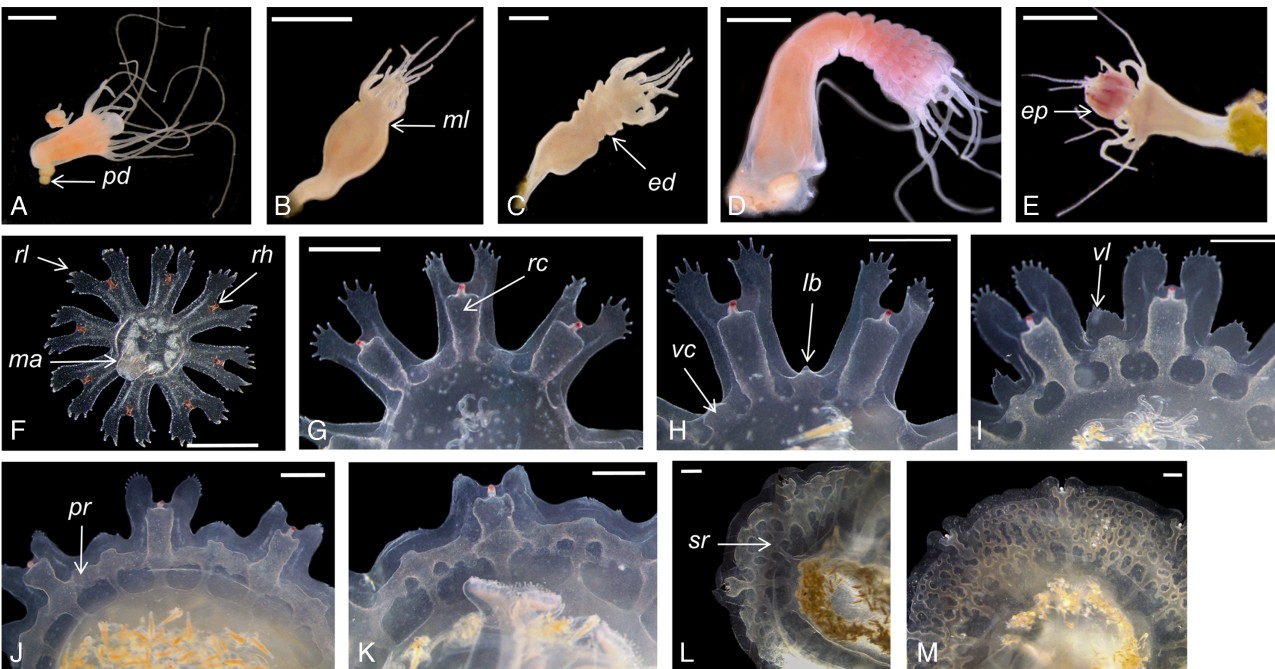

**Figure 2** *Catostylus tagi* life cycle from scyphistoma (A–E) to metaephyra and development stages of gastric system (F–M). *ed*, ephyrae disc; *ep*, ephyrae; *lb*, lappet bud; *ma*, manubrium; *ml*, marginal lobe; *pd*, podocyst; *pr*, primary ring canal; *rc*, rhopalial canal; *rl*, rhopalar lappet; *rh*, rhopalium; *sr*, secondary ring canal; *vc*, velar canal; *vl*, velar lappet. *Scale bars*: all one mm except F: 500 µm. (Photo credit Sonia KM Gueroun).

stage of strobilation, the calyx elongated and the first marginal lobe formed *via* constriction of the upper part of the calyx (Fig. 2B); additional, marginal lobes progressively formed beneath this initial one (Fig. 2C). The lappets and rhopalii appeared (Fig. 2D) and the scyphistoma tentacle progressively achieve complete resorption. After release of the final ephyra, the residuum developed new tentacles (*n* = 16) and hypostome (Fig. 2E).

Newly released ephyrae (stage 0) typically had eight lappets with a pair of antler palm-like rhopalial lappets with two to seven finger-like appendages (Fig. 2F). There was one rhopalium per lappet. The tips of the rhopalial canals ended at the red-coloured rhopalium base; however, the eight velar canals were either not developed or undistinguishable. The eight rhopalial canals were slightly forked and with rounded points. There were one or two gastric filaments per quadrant. The manubrium presented a four-lipped shape (Fig. 3A). Ephyrae exhibited colours from dark pink to dark red. Measurements of newly released *C. tagi* ephyrae are summarized in Table 1.

As the timing of the developmental stages of ephyrae are affected by environmental conditions (*e.g.*, temperature, feeding), the stages are listed below in chronological order without designated time period:

*Stage 1* (Fig. 2G): The TBD doubled to 4.8 ± 0.4 mm. Eight rhombical velar canals appeared. The velar canal thickened and the tips rounded-up. The first oral tentacles developed on the distal ends of the manubrium (Fig. 3B).

**Table 1  Body proportions of *C. tagi*. scyphistoma and ephyra.**

| | |
|---|---|
| **Scyphistoma** | |
| TBL | 1438 ± 240 µm |
| StL | 307 ± 147 µm |
| CL | 1131 ± 190 µm |
| HL | 321 ± 67 µm |
| MDD | 637 ± 120 µm |
| StL% TBL | 21% |
| CL % TBL | 79% |
| HL% TBL | 23% |
| MDD% CL | 57% |
| **Ephyra** | |
| TBD | 2514 ± 426 µm |
| CDD | 968 ± 166 µm |
| TMLL | 822 ± 143 µm |
| RLL | 415 ± 79 µm |
| LStL | 418 ± 105 µm |
| CDD% TBD | 39% |
| TMLL% TBD | 33% |
| RLL% TBD | 17% |
| RLL% TMLL | 51% |
| LStL% TMLL | 51% |
| TMLL% CDD | 119% |

*Stage 2* (Fig. 2H): Lappet bulbs grew between the marginal lappets. The velar canals formed a pair of branches midway. The manubrium started to split into four oral arms (Fig. 3C).

*Stage 3* (Fig. 2I): The lappet bulbs developed into serrated velar lappets. The velar canals lengthened centrifugally and the rhopalial canals formed a pair of side branches that grew centrifugally toward the velar canals. The four oral arms divided to form eight arms (Fig. 3D).

*Stage 4* (Fig. 2J): The side branches of the velar canals fused with the pair of side branches of the rhopalial canals to form a primary ring canal. The velar lappets extended outward while their serrations retracted.

*Stage 5* (Fig. 2K): The velar lappets continued their extension outwards. The serrations on the antler palm-like rhopalial lappets retracted. Two new canals grew on the velar canal, developing centripetally (tertiary canals) parallel to the radial canals, while two additional canals developed horizontally toward the rhopalial canal.

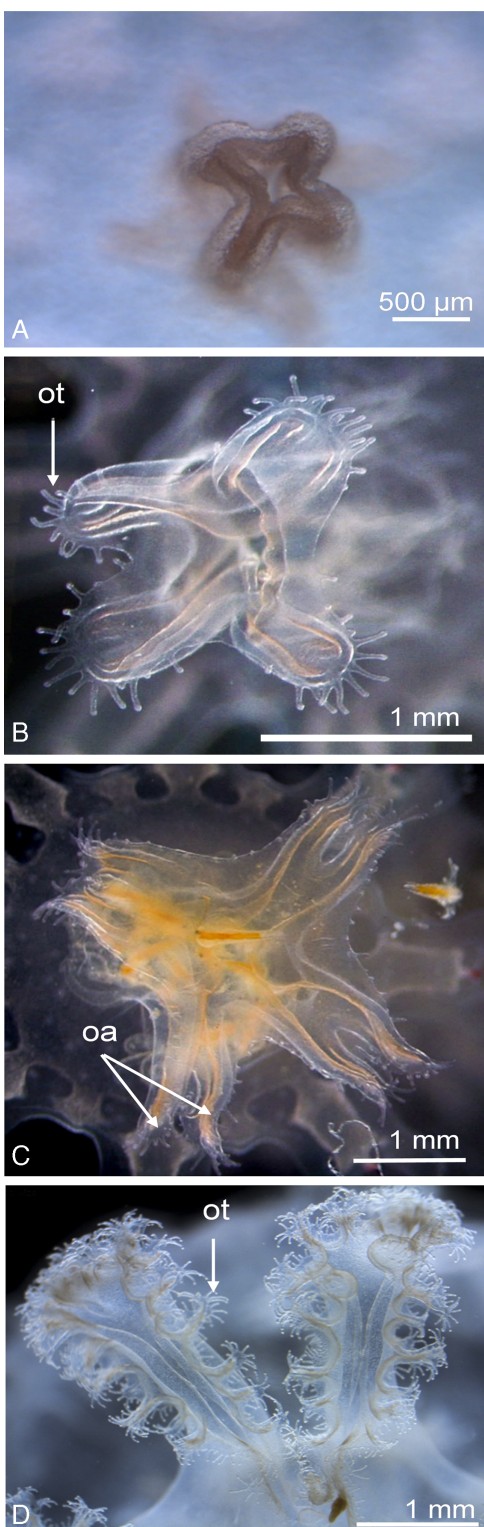

**Figure 3 Enlarged view of the mouth development of *Catostylus tagi*.** (A) Cross-shaped mouth without oral tentacles of a stage 0 ephyrae. (B) Appearance of tiny oral tentacles at the lips mouth in stage 1. (C) Oral lips distally divided to eight oral arms in stage 2. (D) A total of two oral arms in stage 6. *oa*, oral arm; *ot*, oral tentacles. (Photo credit Sonia KM Gueroun).

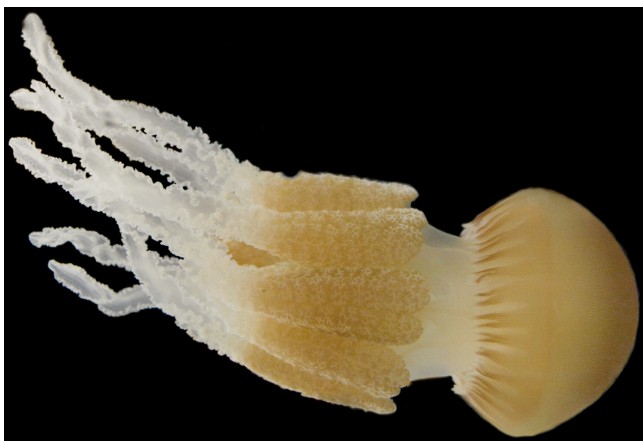

**Figure 4 Photography of a fully developed *Catostylus tagi* medusae reared in the Lisbon Oceanário (Photo credit Raul Gouveia).**

*Stage 6* (Fig. 2L): The midway-side branches of the velar canal fused with the radial velar canals to form a secondary ring canal. The centripetal canal fused with the second ring and continued growing centripetally. Below the second ring, the radial canal developed another set of side branches to form the final ring canal. The velar lappet extremities extended until reaching the rhopalial lappets yo complete the umbrella.

*Stage 7* (Fig. 2M): The tips of the centripetally growing canals avoid the fusion, instead protruding into the space between the last ring canal and the stomach. The floor and the roof of the canals fused, forming woven "*Inseln*", constituting a mesh network of anastomosing canals.

A total of 5 months after ephyrae are released from the strobilae, juvenile *C. tagi* are fully developed (Fig. 4).

## Temperature and salinity effect on planulae development

Planulae of *C. tagi* showed relatively low mortality (≤20 %), especially at 15 °C. No mortality was observed at 30‰ salinity (Fig. 5). The first planulae settlement occurred within four to six hours in all treatment combinations. The final settlement proportion varied from 53% (15 °C and 25‰) to 100% (25 °C and 25–35‰). The planktonic duration (Fig. 6, Table 2) was significantly influenced by temperature ($p < 0.001$) and salinity ($p < 0.01$), however, no interaction was detected ($p = 0.31$). Tukey *post hoc* testing revealed planula settlement to be faster at higher temperatures (25 °C), while the planktonic stage duration was significantly prolonged at 30‰ salinity when compared to 35‰.

No polyps with tentacles were observed at 15 °C, all salinities considered. Polyp development was enhanced at higher temperatures, being optimal at 25 °C for all salinities. Up to 54.2% of the planulae developed into polyps under conditions of 25 °C and 20‰. During the experiment, polyps developed a maximum of eight tentacles, seen at the highest temperature (25 °C) with lower salinities (20 and 25‰). Morphological deformities were not detected.

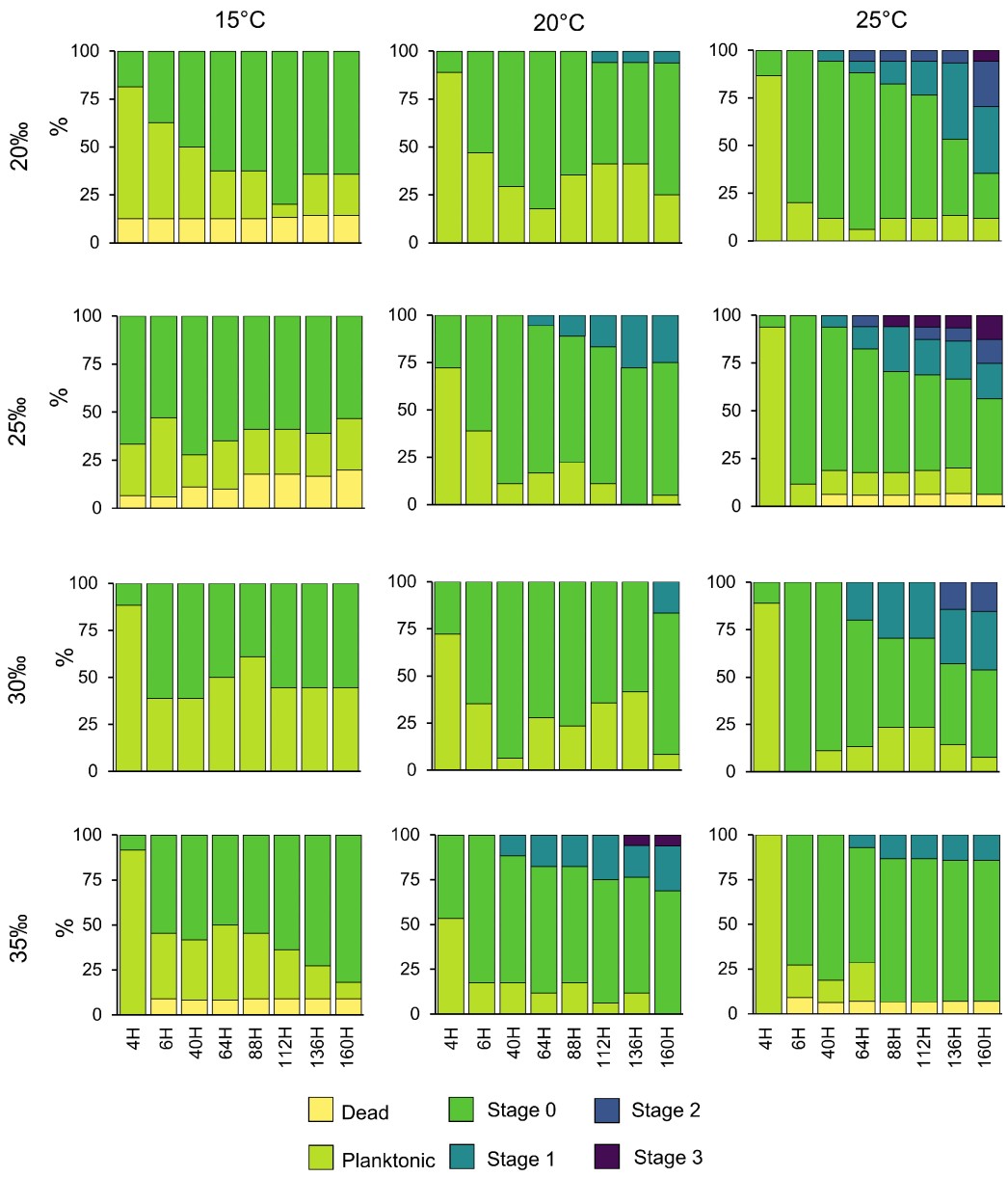

**Figure 5** *Catostylus tagi* **planula survival and development in different temperature and salinity regimes.** Stage 0: Settled; Stage 1: 1–4 tentacles; Stage 2: 5–7 tentacles; Stage 3: 8–16 tentacles.

## Temperature and food effect on asexual reproduction

Podocyst production, which varied between 0.8 ± 0.9 and 1.7 ± 1.5 podocysts per scyphisotma, was neither influenced by temperature ($p$ = 0.21) nor by feeding regime ($p$ = 0.3), with no significant interaction detected ($p$ = 0.17) (Fig. 7, Table 3).

At 20 and 15 °C, strobilation occurred only in three groups: 20 °C-$R_{group}$ (61.1%), 20 °C-$A_{group}$ (61.1%) and 15 °C-$A_{group}$ (28%). *Pre-str*, which was significantly influenced by temperature and diet, was shorter in the 20 °C-$R_{group}$ (11 ± 4 days) and 20 °C-$A_{group}$ (15 ± 7 days) than in the 15 °C-$A_{group}$ (32 ± 3 days) (Tables 2, 3).

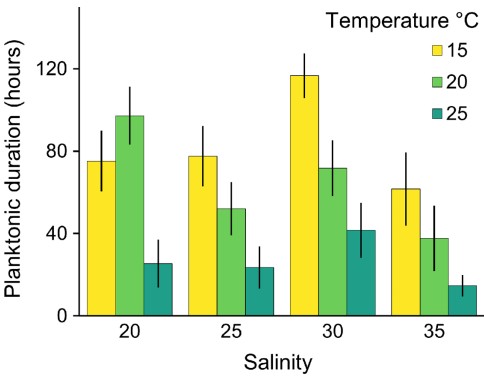

**Figure 6** Average (± SD) of the planktonic duration of *Catostylus tagi* planulae exposed to different temperature and salinity regimes.

**Table 2 Statistical results on the effects of temperature, salinity and diet on *C. tagi* different life stages (ZIqP = Zero-inflated quasi-Poisson model).**

| Variable tested | Planktonic duration (ANOVA) | Podocyst (ANOVA) | Bet-str (Wilcox) | Str (*t*-test) | Ephyra (*t*-test) | Pre-str (ZIqP) | Estimate | Std. Error | z value | p value |
|---|---|---|---|---|---|---|---|---|---|---|
| Temperature | $F (2, 18) = 25.8$ | $F (1, 18) = 1.62$ | – | – | – | *Count model* | | | | |
| | $p < 0.001$ | $p = 0.21$ | | | | Intercept | 6.17 | 0.47 | 12.99 | <0.001 |
| Salinity | $F (3, 18) = 4.1$ | – | – | – | – | Temperature | −0.16 | 0.03 | −5.37 | <0.001 |
| | $p < 0.01$ | | | | | Feed | −0.33 | 0.15 | −2.19 | 0.03 |
| Feed | – | $F (2, 18) = 1.23$ | $p = 0.8$ | $p < 0.01$ | $p = 0.7$ | *Zero-inflated model* | | | | |
| | | $p = 0.3$ | | | | | | | | |
| Temperature X salinity | $F (6, 18) = 1.21$ | – | – | – | – | Intercept | 11.79 | 2.49 | 4.74 | <0.001 |
| | $p = 0.3.1$ | | | | | Temperature | −0.51 | 0.12 | −4.15 | <0.001 |
| Temperature X feed | – | $F (1, 18) = 1.89$ | – | – | – | Feed | −1.37 | 0.37 | −3.71 | <0.001 |
| | | $p = 0.17$ | | | | | | | | |

Ephyrae release was only observed at 20 °C. The duration of *bet-str* ($R_{group}$: 5.7 ± 1.4 days; $A_{group}$: 5.6 ± 2.6 days) and the number of ephyrae produced by strobilation ($R_{group}$: 6.3 ± 3.5; $A_{group}$: 6.9 ± 3.9) were not significantly affected by the diet (Tables 2, 3). The strobilation period was significantly shorter in the $R_{group}$ (2.5 ± 2.3 days) than in the $A_{group}$ (26.4 ± 3.9 days) (Tables 2, 3). Twenty-seven per cent of the scyphistomae in the 20 °C-$R_{group}$ were able to perform new strobilation 10 to 16 days after ending the first strobilation. The number of ephyrae produced by those scyphistomae did not change between the first (4 ± 1.7) and the second strobilation (3.7 ± 0.6).

## Seasonal distribution of *Catostylus tagi* adults in the Tagus estuary

During 2019, sightings of adult *C. tagi* were reported *via* the GelAvista smartphone App along both margins of the Tagus estuary, from the inner bay and Cala do Norte to the

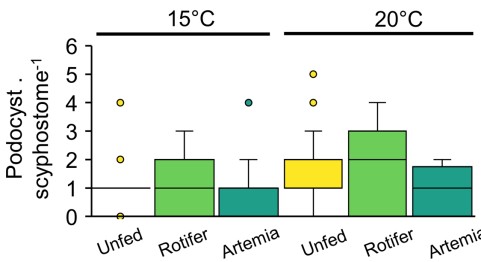

**Figure 7 *Catostylus tagi*** average (±SD) podocyst production per scyphistoma under different temperature and diet regimes.

**Table 3 *Catostylus tagi*** ephyrae production and *Pre-str*, *bet-str* and *str* duration in day (±SD) for each of the treatment groups. (*n*): number of scyphistoma.

| | 15 °C | | | 20 °C | | |
|---|---|---|---|---|---|---|
| | Unfed | Rotifers | Artemia | Unfed | Rotifers | Artemia |
| Ephyra. strobilation$^{-1}$ | – | – | – | – | 6.3 ± 3.4 (13) | 6.9 ± 3.6 (9) |
| Strobilation timing (day) | | | | | | |
| Pre-strobilation | – | 32.2 ± 3 (5) | – | – | 10.6 ± 5.4 (14) | 15 ± 6.6 (11) |
| Bet-strobilation | – | – | – | – | 5.7 ± 1.3 (13) | 5.6 ± 2.2 (10) |
| Strobilation | – | – | – | – | 2.5 ± 2.5 (13) | 6.4 ± 3.7 (10) |

estuary's opening to the Atlantic Ocean (Fig. 8A). Sightings were recorded in all months of the year except April (Fig. 8B). Many sightings (>5) were recorded from September to February, a proxy for a higher abundance of this species.

## DISCUSSION

This work represents the first complete description of the life cycle of *C. tagi*, including the first insights into its ecology. We emphasise that baseline studies on important blooming species, such as the current study, are a crucial aspect of understanding jellyfish fluctuations and their economic impact.

The jellyfish *C. tagi* displays a typical metagenetic life cycle for Rhizostomida, including a benthic scyphistoma phase that reproduces asexually *via* strobilation, releasing ephyrae that grow into pelagic medusae that reproduce sexually. Our observations of adult gonads from individuals (18 medusae, six used in the present study) collected in the Tagus estuary (Portugal) for the fertilisation showed the absence of planulae. Several scenarios may explain this absence; either *C. tagi*, unlike *C. mosaicus* (*Pitt, 2000*) and some other Rhizostomida (Table 4), is a non-brooding species or *C. tagi* sexual reproduction occurs later in the year than the present sampling period. Additional studies on extended periods are necessary to clarify this issue.

Adult medusae of *C. tagi* and *C. mosaicus* present multiple clear morphological distinctions (*Jarms & Morandini, 2019*), such as in the umbrella (hemispherical and up to 35 cm wide with coarse granulation for *C. mosaicus* while flattened hemispherical and up to 65 cm wide with fine granulation for *C. tagi*), the number of total marginal lappets

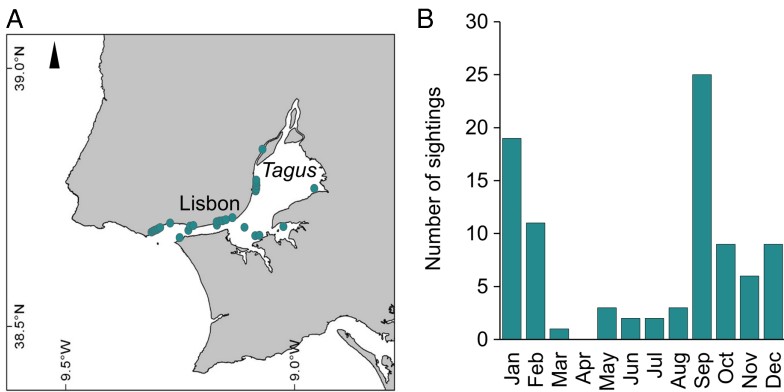

**Figure 8** Spatial (A) and monthly (B) occurrence of *Catostylus tagi* medusa along the Tagus estuary in 2019.

(128 for *C. mosaicus* and 80 *C. tagi*); the velar lappet shape (oval in *C. mosaicus* but triangular in *C. tagi*) or the terminal portion of tapering filaments on the oral mouth and the purple margin on the umbrella only seen in *C. tagi*. In contrast, distinguishing between earlier stages of these species is more of a challenge (Table 4). Comparisons between both the polyps and ephyrae of *C. tagi* and *C. mosaicus* showed very few differences between the species. Both ephyra species develop into an eight-rayed medusa with 16 antler palm-like rhopalial lappets and have similar body proportions (*Straehler-Pohl & Jarms, 2010*). The main contrast between these two species lies in the velar canal shape, which is rhombical in *C. tagi* and spade-like in *C. mosaicus* (*Straehler-Pohl & Jarms, 2010*). These anatomic characteristics of *C. tagi* ephyra distinguish the species from the other two Rhizostomatidae occurring in the same geographical area, *Rhizostoma luteum* (rhombical shape velar canal and slightly forked rhopalial canal) (*Kienberger et al., 2018*) and *Rhizostoma octopus* (flat rhombical shape velar canal and slightly forked rhopalial canal) (*Holst et al., 2007*). Traditionally, the Discomedusae order was divided into two orders: the Semaeostomeae and the Rhizostomeae, including the Kolpophorae (Cepheida) and the Daktyliophorae (Rhizostomida) suborders. Recently, *Jarms & Morandini (2019)* proposed distinguishing three suborders: Semaeostomeae, Cepheida and Rhizostomida. The distinction between the Cepheida and Rhizostomida is based on development of the gastrovascular network; gastric system develops centrifugally in the Cepheida and centripetally in the Rhizostomida (*Holst et al., 2007*; *Straehler-Pohl, 2009*). *Catostylus tagi* gastric development followed the same model of the previously described Rhizostomida such as *Rhizostoma pulmo* (*Fuentes et al., 2011*) and *Stomolophus* sp.2 (*Gómez-Salinas, López-Martínez & Morandini, 2021*).

Scyphozoan species exhibit several propagation strategies, including various budding modes (*e.g.*, lateral budding, budding from the stolon and motile bud-like tissue particles) and podocysts (*Arai, 1997*). These propagation strategies are species-specific. Some species adopt a mono-mode, such as free-swimming planuloids in *Phyllorhiza punctata* (*Rippingale & Kelly, 1995*) or podocysts in *Rhopilema nomadica* and *Rhizostoma luteum* (*Lotan, Ben-Hillel & Loya, 1992*; *Kienberger et al., 2018*). In contrast, some species combine

**Table 4 Polyp and ephyra morphology of Rhizostomida species. *rc*: rhopalial canal, *vc*: velar canal, \*: brooding species.**

| Species | Polyp | | | | | Ephyrae | | | | | | Source |
|---|---|---|---|---|---|---|---|---|---|---|---|---|
| | Polyp size range (mm) | MDD (mm) | Nb. of tentacles | Scyphistome, hypostome | Asexual reproduction | Strobilation (ephyrae per strobilation/ strobilation type) | Ephyrae size after release (mm) | Nb. of marginal lappets | Shape of rhopalial lappet | Shape of velar canal/ rhopalial canal | Ephyra colour | |
| *Catostylus tagi* | 1.08–1.83 | 0.44–0.86 | 16 | Long, club-shaped | Podocysts | 1, monodisk (rare) Up to 15, polydisk | 1.5–3.1 | 8 | Antler palm-like, with 2 to 7 finger-like appendages | Rhombical/ slightly forked, rounded points | Dark pink to dark red, red statocysts | Present study |
| *Catostylus mosaicus*\* | 1.57–1.90 | 0.69–0.81 | 12–20 | Long, club-shaped | Lateral polyp buds, podocysts, pedalocysts, longitudinal fission | 1, monodisk 2–5, polydisk | 1.9 – 2.26 | 8 | Antler palm-like, with 3 to 5 finger-like appendages | Spade-like *vc*/ slightly forked *rc* | na | 1, 2, 3 |
| *Rhizostoma luteum*\* | 1.34–2.5 | 1.02 | 14–16 | Conspicuous and flexible in all stage | Podocysts | 1, monodisk | 3.41–4.52 | Typical 8, 11 | Bread knife shaped | Rhombical *vc*/ slightly forked *rc* | Light yellow to light brown | 4 |
| *Rhizostoma pulmo* | 0.96–2.15 | 0.53–1.16 | 14–16 | Long, club-shape and flexible | Lateral polyp buds, podocysts, lateral buds, stolonial polyp buds, pedalocysts | 8–13.5, polydisk, oligodisk | 2.28–3.93 | Typical 8, 5–9 | Spade like to lancet shaped | Rhombical or absent *vc*/ slightly forked *rc* | Milk transparent to opaque white | 2, 3, 5, 6, 7 |
| *Rhizostoma octopus* | 1.9–2.3 | 1.25 | 16–24 | Long and flexible | Podocysts, lateral buds, longitudinal fission | 1, monodisk Up to 15, polydisk | 2 7–5.96 | 8 | Bread knife-like | flat rhombical *vc*/slightly forked *rc* | Milky transparent, light yellow to light brown | 2, 8, 9 |
| *Rhopilema esculentum* | 1.00–3.50 | 1.6 | 16 | Moderately long | Podocysts | 7–17, polydisk | 1.5–4.0 | 8 | Talon-shaped with 4–6 branches ; Hand saped with 4–6 finger-like appendages | arrow tip-like *vc*/spatula-like *rc* | Milky to transparent | 2, 3, 10 |

(Continued)

| Species | Polyp | | | | | Ephyrae | | | | | | Source |
|---|---|---|---|---|---|---|---|---|---|---|---|---|
| | Polyp size range (mm) | MDD (mm) | Nb. of tentacles | Scyphistome, hypostome | Asexual reproduction | Strobilation (ephyrae per strobilation/ strobilation type) | Ephyrae size after release (mm) | Nb. of marginal lappets | Shape of rhopalial lappet | Shape of velar canal/ rhopalial canal | Ephyra colour | |
| *Rhopilema nomadica* | 1.8–2.0 | na | 16 | Large clavate shape, third of polyp length | Podocysts | 5–6, polydisk | 1.5–2.0 | 8 | Single or twin-typed, lancet-shaped | Convex with arched corners | na | 11 |
| *Rhopilema verrilli** | 2.5 | 0.35 | 8–20 | Large, flexible, quadrate, irregular in outline | Podocysts, pedalocysts | 1, monodisk Up to 3, polydisk | 3.0 | 8 | Rounded, slender, pointed distally | Rhopalar pouches with prominent «horns» | Peach, orange-red to rose coloured ; birefringent, bright yellow gold statocysts | 12, 13 |
| *Lychnorhiza lucerna* | 1.5 | 0.55–0.8 | 18–22 | Prominent dome-shaped | Podocysts | 3, polydisk | 1.4 | 8 | Hand shaped with 2 to 9 tips | na/square-shaped ends with slight lateral horns rc | Translucent | 14 |
| *Nemopilema nomadica* | 2.6 | 0.8–1.1 | 16 | Dome-shaped, one third of scyphistome height | Podocysts | 3–7, polydisk | 2.2–3.8 | 8 | Hand shaped with 2 to 6 pointed tips | Unforked triangular vc/unforked, spatula shaped rc | Translucent | 15 |
| *Stomolophus meleagris* | 2.0 | na | 16 | Large, flexible, and dome-or knob-shaped | Podocysts | 1, monodisk 2-3, polydisk | 1.5–2.0 | 8 | Slender, distally pointed | Adradial bulges vc/ blunt-ended rc | Pale straw coloured | 16 |
| *Cephea cephea** | 1.4–2.9 | 0.44–0.6 | 14–17 | Short, club-shaped | Lateral budding, swimming buds | 1, monodisk | 1.6–3.24 | 8 | Round spoon-shaped | Rhombical vc / slightly forked rc | Pale yellow to yellowish-brown | 2, 3, 17 |
| *Cotylorhiza tuberculata** | 3.23–5.0 | 0.82–0.86 | 16–17 | Short, cylindrical | Lateral budding, swimming buds | 1, monodisk | 1.5–3.25 | 8 | Rounded to rounded spoon shaped | spade-like to slightly rhombical vc/slightly forked rc | Transparent with yellow hemmed gastric system | 2, 3, 18, 19 |

| Species | Polyp | | | | | Ephyrae | | | | | | Source |
|---|---|---|---|---|---|---|---|---|---|---|---|---|
| | Polyp size range (mm) | MDD (mm) | Nb. of tentacles | Scyphistome, hypostome | Asexual reproduction | Strobilation (ephyrae per strobilation/ strobilation type) | Ephyrae size after release (mm) | Nb. of marginal lappets | Shape of rhopalial lappet | Shape of velar canal/ rhopalial canal | Ephyra colour | |
| *Mastigias papua** | 1.0–10.22 | 0.4–0.92 | 15–18 | Very short, cylindrical | Planuloids | 1, monodisk | 1.5–3.91 | 8 | Rounded; Trapered, broad spoon shaped | spade-like to slightly rhombical *vc*/slightly forked *rc* | Brown to orange brown | 2, 3, 20, 21, 22 |
| *Phyllorhiza punctata** | na | na | 16 | na | Ciliated buds | 1, monodisk | 0.46–2.5 | 8 | Pointed spoon shaped | spade-like to slightly rhombical *vc*/slightly forked *rc* | Yellowish brown to ochre | 3, 23 |
| *Cassiopea andromeda** | 4.72–10.0 | 1.95 | 32 | Conspicuous long, tetragonal | Swimming buds, planuloids | 1, monodisk | 3.69–3.95 | 12–23 | Spatula-like | tongue-like and reach the tips of the velar lappets, tips are rounded *vc*/long forked *rc* | Yellowish green | 2, 3, 24, 25 |
| *Cassiopea xamachana* | na | 1.5–2.0 | Up to 42 | na | Planula-like larvae | 1, monodisk | na | na | na | na | na | 26 |

**Note:**

Source: 1: Pitt (2000); 2: Straehler-Pohl (2009); 3: Straehler-Pohl & Jarms (2010); 4: Kienberger et al. (2018); 5: Fuentes et al. (2011); 6: Purcell et al. (2012); 7: Schiariti et al. (2014); 8: Holst et al. (2007); 9: Holst & Jarms (2007); 10: You et al. (2007); 11: Lotan, Ben-Hillel & Loya (1992); 12: Cargo (1971); 13: Calder (1973); 14: Schiariti et al. (2008); 15: Kawahara et al. (2006); 16: Calder (1982); 17: Sugiura (1966); 18: Kikinger (1992); 19: Prieto et al. (2010); 20: Sugiura (1964); 21: Sugiura (1963); 22: Sugiura (1965); 23: Rippingale & Kelly (1995); 24: Gohar & Eisawy (1960a); 25: Gohar & Eisawy (1960b); 26: Bigelow (1900).

two or more strategies, as seen in *Aurelia* spp. (*e.g.*, lateral budding, lateral budding through stolons, reproduction from parts of stolons/stalks, motile bud-like tissue particles and podocysts) (*Schiariti et al., 2014*). Unlike *C. mosaicus*, which combines various strategies, including lateral polyp buds, podocysts, pedalocysts and longitudinal fission (*Pitt, 2000*; *Straehler-Pohl, 2009*), *C. tagi* propagation seems to be limited to podocysts, at least under the tested conditions. Moreover, podocysts production was not influenced by temperature (15–20 °C) nor feed regime (unfed, rotifer or *Artemia*). The reproductive strategy adopted by a species of scyphozoan plays a significant role in the polyp reproduction rate (*Schiariti et al., 2014*) and, consequently, the potential for medusae outbreaks. Among the various propagation modes, podocysts present the lowest reproduction rate. Podocysts contain stored nutritional reserves in carbohydrates, lipids, and proteins (*Black, 1981*; *Chapman, 1968*), which can remain dormant for an extended period. *Thein, Ikeda & Uye (2012)* found that *Aurelia aurita* s.l. podocysts were able to survive for up to 3.2 years. Early studies speculated that podocysts represents an asexual reproduction strategy induced by poor environmental conditions and protection against predators (*Cargo & Schultz, 1967*). For instance, starvation or below food threshold conditions trigger podocyst formation in *Aurelia aurita* s.l. (*Han & Uye, 2010*; *Thein, Ikeda & Uye, 2012*). However, results in several other species have shown that podocyst formation is enhanced by increasing food supply and temperature (*Kawahara, Ohtsu & Uye, 2013*; *Schiariti et al., 2014*). This contrast is related to the asexual reproduction mode (*e.g.*, mono-mode or multi-mode) adopted by the species. In mono-mode species (*e.g.*, *Lychnorhiza lucerna, R. pulmo, Rhopilema esculentum, Nemopilema nomurai* and *Stomolophus meleagris*) in which podocysts are the only form of asexual polyp reproduction besides strobilation, increase in temperature and food supply enhance podocyst production; meanwhile, these same factors boost budding and stolon production in multi-mode species. Although *C. tagi* only produced podocysts during the present study, it would be premature to categorically consider *C. tagi* as a mono-mode species. More investigations will be required with longer trials analysing a wider range in temperatures and the effect other factors, such as salinity, oxygen and pH.

In line with other Rhizostomida species (with the exception of *R. luteum*), polydisc was the predominant strobilation type observed in *C. tagi* (Table 4). The first scyphistomae were observed 3 days after fertilisation and strobilated approximately 10 days later (18 °C, 35 ‰). Strobilation onset, duration, and the number of ephyrae produced by polydisc scyphistoma are influenced by environmental factors, such as temperature (*Purcell et al., 2012*) and food supply (*Wang et al., 2015*). Other factors, such as age and size (*Russell, 1970*; *Holst, 2012*), are intrinsic to scyphistoma. Under the various experimental conditions used to rear *C. tagi* in the current study, strobilation onset was faster at higher temperatures and was triggered by the availability of food (no strobilation in unfed scyphistoma). Surprisingly, while diet type can affect the number of produced ephyrae (*Purcell et al., 1999*; *Wang et al., 2015*), *C. tagi* ephyrae production was not influenced by diet type (rotifer or *Artemia*), indicating that both diets are suitable.

Under the diverse conditions (temperature and diet) used in the present study, including broodstock and the experiments, *C. tagi* scyphistomae underwent repeatedly

strobilated twice a month (≥18 °C), producing up to 15 ephyrae (6.8 ± 3.6); this ephyrae production exceeds reported values for *C. mosaicus* of up to 5 ephyrae per strobilation (*Pitt, 2000*). However, it is unknown whether the culture conditions of the previous *C. mosaicus* study were optimum for strobilation. *C. tagi* scyphistoma strobilated through a wide temperature range varying from 15 to 25 °C (Gueroun, *personal communication*). This temperature range (15–25°C) has been recorded in the Tagus (*Gameiro, Cartaxana & Brotas, 2007*), which may explain the continuous occurrence of *C. tagi* medusae in the estuary from May to March.

In the scyphozoan life cycle, the lecithotrophic and short-lived (around 10 days) planula larval stage is undoubtedly the most vulnerable life stage. The recruitment of a new generation depends upon larvae pre-settlement survival, as well as their capacity to settle and metamorphose into the robust feeding polyp stage before their energy reserves run out. The Tagus estuary displays high spatial and temporal hydrographic fluctuations (*Gameiro, Cartaxana & Brotas, 2007*; *Rodrigues et al., 2017*). The data collected on *C. tagi* medusa presence in the Tagus estuary between 2016 and 2019 (*GelAvista, 2021*; present study) shows the occurrence of the species from May to March (2019), suggesting a wide tolerance of the medusa stage to varying temperatures and salinities, similar to previous observations in *C. mosaicus* (*Loveridge et al., 2021*). In our experiment, the euryhalinity and eurythermal characteristics of the planula stage of *C. tagi* were confirmed, as results demonstrated a low mortality rate and faster metamorphosis into polyps at higher temperatures. The euryhalinity exhibited by *C. tagi* planulae might also be retained by polyps, as the species-specific tolerance to low salinities was largely comparable for both planulae and polyps in several estuarine and brackish water scyphozoan species, including *Chrysaora pacifica*, *N. nomurai*, *R. esculentum* and *Aurelia coerulea* (*Holst & Jarms, 2010*; *Conley & Uye, 2015*; *Dong et al., 2015*; *Takao & Uye, 2018*; *Feng et al., 2018*).

The timing and magnitude of scyphomedusae recruitment depends on numerous factors, including the survival and settlement success of planulae, the abundance and rate of polyp strobilation and the rate of ephyrae survival; notably. All of these factors are influenced by various environemental conditions, mainly temperature, salinity and diet. Taking into account the spatial-temporal distribution of *C. tagi* in the Tagus estuary (May to March), the wide temperature and salinity tolerance of planulae, and the strobilation temperature range (15 to 25 °C), the possibility of several successive cohorts is suggested, as opposed to a longevity of 11 months in the medusa stage. The absence of medusae in April may be explained by a recruitment cessation resulting from minimal strobilation or low ephyrae survival rates, during the coolest months of November and December (≤12 °C). Strobilation cessations or slowdowns during low temperature periods have been documented and have been attributed to diminished abilities of polyps to feed or increased time required for strobilation (*Widmer, Fox & Brierley, 2016*; *Purcell et al., 2012*)

## CONCLUSION

The present observations indicate a high tolerance and plasticity of the species, contributing to a better understanding of the biology and ecology of *C. tagi*. The euryhaline and eurytherm characteristics displayed by the planulae and reflected by the temporal

and spatial distribution of medusae in the Tagus estuary are advantageous for future *C. tagi* aquaculture, research and production. Further studies must be conducted on the polyp and ephyra stages to determine critical environmental factors affecting asexual reproduction and growth. Finally, studies such as the one we present here are essential for evaluating the response of *C. tagi* to climate change, as well as predicting any temporal and geographic spreading of the species.

## ACKNOWLEDGEMENTS

The authors are grateful to the Curator Núria Baylina and the aquarists' team (Raul Gouveia, Catarina Barraca and Carlos Cunha) of the Oceanário de Lisboa for facilitating the sampling inside the dock and for providing us space and the conditions within which we could conduct our present research. The authors are also grateful to Susana Garrido from IPMA for the support and the citizens participating in the Gelavista project. We thank the Editor and three Referees for their constructive criticism and valuable observations that greatly strengthened the original manuscript.

### Funding

This work was supported by the project GoJelly—A gelatinous solution to plastic pollution—funding from the European Union's Horizon 2020 research and innovation programme under grant agreement No. 774499. This study also had the support of Fundação para a Ciência e Tecnologia (FCT), through the strategic project [UIDB/04292/2020] granted to MARE UI&I. João Canning-Clode is funded by national funds through FCT–Fundação para a Ciência e a Tecnologia, I.P., under the Scientific Employment Stimulus-Institutional Call-[CEECINST/00098/2018]. This research was also supported by the GelAvista citizen Science program under the Project PLANTROF Dinâmica do plâncton e transferência trófica: Biodiversidade e ecologia do zooplankton de Portugal: Mar 2020—Programa Operacional Mar 2020 Portaria N. 118/2016. There was no additional external funding received for this study. The funders had no role in study design, data collection and analysis, decision to publish, or preparation of the manuscript.

### Grant Disclosures

The following grant information was disclosed by the authors:
European Union's Horizon 2020 research and innovation programme: 774499.
Fundação para a Ciência e Tecnologia (FCT): UIDB/04292/2020.
Fundação para a Ciência e a Tecnologia: CEECINST/00098/2018.
Project PLANTROF Dinâmica do plâncton e transferência trófica: Biodiversidade e ecologia do zooplankton de Portugal: 118/2016.

### Competing Interests

Antonina dos Santos is an Editor for PeerJ.

## Author Contributions

- Sonia K.M. Gueroun conceived and designed the experiments, performed the experiments, analyzed the data, prepared figures and/or tables, authored or reviewed drafts of the paper, and approved the final draft.
- Tatiana M. Torres performed the experiments, authored or reviewed drafts of the paper, and approved the final draft.
- Antonina Dos Santos analyzed the data, authored or reviewed drafts of the paper, and approved the final draft.
- Nuno Vasco-Rodrigues performed the experiments, authored or reviewed drafts of the paper, and approved the final draft.
- João Canning-Clode analyzed the data, authored or reviewed drafts of the paper, and approved the final draft.
- Carlos Andrade analyzed the data, authored or reviewed drafts of the paper, and approved the final draft.

## Data Availability

The raw data and the R code script are available in the Supplemental Files.

## Supplemental Information

Supplemental information for this article can be found online at http://dx.doi.org/10.7717/peerj.12056#supplemental-information.

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
