# Peer review of "Catostylus tagi (Class: Scyphozoa, Order: Discomedusae, Suborder: Rhizostomida, Family: Catostylidae) life cycle and first insight into its ecology"

_PeerJ, doi:10.7717/peerj.12056_

## Round 0.1 · original submission · Major Revisions

Dear Dr Gueroun,

We have received the reports from our reviewers on your manuscript.

Based on the advice received, the Editor feels that your manuscript could be accepted for publication should you be prepared to incorporate major revisions. When preparing your revised manuscript, you are asked to carefully consider the reviewers' comments which are attached, and submit a list of responses to the comments. Your list of responses should be uploaded as a file in addition to your revised manuscript.

Best regards,
Alexander Ereskovsky

·

Basic reporting

As a not native English speaking person I cannot appreciate true value of the language, but my opinion is the English can be improved.
The rest is Ok, except the figures, some of which need editting in lebels.

Experimental design

No comment - everithing is Ok

Validity of the findings

Everything is Ok, but discussion should be improved.

Additional comments

ABSTRACT
The last paragraph of the abstract describes the methodology and results in short, excessively short. It must present more results in more details.

In the Results it is a long and detailed description of the ephyra development (growth) stages, missing the temporal parameters. And no discussion of these data in Discussion. The authors need either discuss (compare) their data with the similar descriptions of other species, or remove such detailed descriptions.

The final part - the spatial and seasonal distribution of the species (in the Results and Discussion) - in its present description sound poorly connected with the rest of the manuscript. One possibility - is to connect the revealed ecological preferences of the species with its records in the region. At the moment it sounds vice-versa.

lines 1, 38, 42, 348, Table 4 - Rhizostomida -you should clearly define the taxonomic classification used in the study. According to WoRMS taxon names are different. Moreover sometimes the spelling is different in different parts of the manuscript: Rhizostomida or Rhizostomidae...

Other comments you will find in 'comments' in the file.

Reviewer 2 ·

Basic reporting

This research, even if it contains original data of undoubted interest, has some criticalities.
English would need to be refined.
References ok.
Article structure should be improved; the morphological / morphometric description is sufficient, but the section concerning the experiments is a bit confusing and unclear.
Fig. 7 is unclear and confusing
There are also some doubts about the experimental design system, see next point.

Experimental design

The authors tested the effect of temperature and salinity on planula development, and this is OK, but why didn't they perform the same type of experiments on the asexual reproductive rate of the polyps?
The authors limited themselves to analyzing the effect of temperature and feeding on asexual reproduction, but this section is somewhat confusing and unclear.
As for the statistical analysis, some doubts fall in the methodological approach adopted. Following the authors’ rationale, the nature of repated measurements (line 194) have not been considered in the models. GLMM or LMM would be appropriate in such type of data. Furthermore, in relation to the performed models, I think that, once a Poisson GLM model was overdispersed, the Authors have to opt for a quasipoisson distribution of error (just changing the overdispersion parameters) since prediceted values (integers) are more realistic than those predicted by a negative binomial distribution.
In general, the whole section describing statistical methods can be improved and Authors choices better justified.

Validity of the findings

I do not doubt that the observations made by the authors in the final discussion are correct, but the problems indicated above make them poorly supported.

Additional comments

I would suggest to refine the research as indicated above, perhaps adding an experimental part on the asexual reproductive rate as a function of the main environmental variables, temperature and salinity, and not only as a function of nutrition. Also review the statistical analysis section, as indicated above.

·

Basic reporting

1) I would strongly recommend this article to be revied by a native speaker. English, which is used here, is ambiguous. Abstract is especially hard to understand due to the fact that sentences are not connected with each other. Moreover, there are multiple typos and some mistakes in terminology.
Problems with language, that I can define:
Lines 1.38.42. “Rhizostomida”, the preferred name of this suborder is Dactyliophorae. Line 1. Also, it is strange to name class and suborder (Scyphozoa, Rhizostomida) without naming the order (Scyphozoa, Rhizostomeae, Dactyliophorae). Maybe naming only class and order will be more appropriate (Scyphozoa, Rhizostomeae).
Lines 32-33 are unreadable. What do you mean by “jellyfish outbreaks are … natural events”? What do you mean by “environmental… activities”? How the first leads to the second?
Line 34. “different jellyfish life stage” – it should be: “different jellyfish life stages”
Line 34.385. “its” – “it’s”
Line 35. “have its full life cycle understood” – I’m not sure the word “understood” is a good fit for the scientific article.
Line 38. “scyphistoma presents 18 tentacles” – The word “presents” doesn’t look like a good choice.
Line 39. “Only podocyst production was observed.” – This sentence should be expanded. In the current state it conflicts with the next sentence, where you talk about strobilation, that you also observed.
Line 41. “ephyrae… at liberation” – The word “detachment” or “release” is usually used.
Lines 52-53. “Tagus estuary” is used twice.
Lines 56-57. What do you mean by “The outbreaks often have severe… ecosystems”? Please rephrase.
Lines 57-60. The sentence is too complicated and thus very hard to understand.
Lines 63.115.157. I would recommend to use the word “attached” instead of “fixed”.
Lines 64-65.224-225.326-327.332-333. “(e.g. budding, swimming buds, podocysts, fission, strobilation) (Arai, 1997)”; “lateral budding by stolon, lateral scyphistoma budding, or pedalocysts”; “various budding modes (e.g. lateral budding, budding from stolon, motile bud-like tissue particles, etc.) and podocysts (Arai, 1997)”; “lateral budding, lateral budding through stolons, reproduction from parts of stolons/stalks, motile bud-like tissue particles, podocysts) (Schiariti et al., 2014)” – The unification of the names of modes of asexual reproduction is required, especially while you citate the same article on this lines.
Line 68. “both phases integral to the study” – I’m not sure the word “integral” fits here.
Lines 112.119.148.149.150.154.271.273.277.280.282.350.366.377.Figure 5.Gigure 6. Please use the sign ‰ , when you specify the salinity.
Line 115. “given to planula choice to” – please rephrase.
Lines 140.230.237-238.249.320.Table 1.Figure 1.Figure 2. “ephyra lappet elongated”; “Ephyrae typically had eight marginal arms with a pair of antler palm-like rhopalial lappets and a single rhopalium per lappet” – unification of the terms is needed. Which part of ephyra do you call “marginal arm”, “marginal lobe”, “lappet”, “marginal lappet”, “rhopalial lappet”, “rhopalia lappet”?
Line 142. “MDD/CL 9 100” – I believe it was supposed to be “MDD/CL x 100”.
Lines 143.144. “CDD/TBD 9 100”, “RLL/TMLL 9 100” – the same.
Line 148. “three temperature” – “three temperatures”.
Line 161. “The effect of temperature and feeding on asexual reproduction” – “The effect of temperature and feeding regime on asexual reproduction”.
Line 162. “two temperature levels” – “two temperatures”.
Line 163. “three feed regimes” – “three feeding regimes”.
Lines 163-164. “Eighteen polyps were tested in each of the six combinations.” – This sentence is ambiguous and thus should be rephrased.
Lines 164-165. The sentence is very hard to understand. Please rephrase.
Line 215-216. 218. “total body length (TBD)” – I believe it was supposed to be “total body length (TBL)”.
Line 218-219. “was distinctly longer than wide” – seems to be a poor choice of words, specifics are required.
Line 227. “strobiles” – “strobilae” is preferable.
Lines 229-230. “calyx upper part” – How do you define “up” and “down” in strobila? It would be correct to use terms “oral”-“aboral”.
Lines 245-248. “The first oral tentacles develop on the distal ends of the manubrium. … The first oral tentacles develop around the manubrium opening” – information seems to be doubled.
Line 258.Table 2. “two other canals develop horizontally” – the word “horizontally” doesn’t describe the position of the structure in the jellyfish body. Please rephrase.
Line 260. The word “canals” is missing.
Line 359. “up to” is used twice.
Table 1. “ep:ephyrae” – “ep:ephyra”.
“ephyrae:500μm” - “ephyra:500μm”
“ephyrae” - “ephyra”
“•18 filiform tentacles •Long, club-shape and flexible” – I believe it was supposed to be “•18 filiform tentacles; long, club-shape and flexible”
“•MDD ≈ 44 % of the TBL” – Is everything correct here? Diameter in relation to length?
Table 2. “Development stages” - “Developmental stages”.
“• Rhopalia canals are sli forked, rounded points • • rhopalial canals slightly forked with rounded points” – information is doubled.
Figure 3. “Enlargement of the mouth development of Catostylus tagi.” – “Enlargement of the mouth of Catostylus tagi.”
“at the lips mouth in (stage 1)” – the content is not clear.
Figure 5. “Stage 1: 1-3 tentacles” – but on the lines 157-158 “Stage 1: 1-4 tentacles”.
Figure 7. “C. tagi asexual reproduction under temperature and diet regimes.” – the word “different” seems to be missing.
“ephyra. strobilation-1” - “ephyrae.strobilation-1”.

2) The background is present, but it lacks the known information on ephyra development. This work is not the first to show the development of gastric system and margin of the Scyphozoan young medusa. For example:
• Holst, S. (2012). Morphology and development of benthic and pelagic life stages of North Sea jellyfish (Scyphozoa, Cnidaria) with special emphasis on the identification of ephyra stages. Marine Biology, 159(12), 2707–2722. doi:10.1007/s00227-012-2028-0
• Holst, S., Sötje, I., Tiemann, H., & Jarms, G. (2007). Life cycle of the rhizostome jellyfish Rhizostoma octopus (L.) (Scyphozoa, Rhizostomeae), with studies on cnidocysts and statoliths. Marine Biology, 151(5), 1695–1710. doi:10.1007/s00227-006-0594-8 (Which is by the way cited in the text).
Lines 61-62. “With some exceptions (e.g. Pelagia noctiluca), Scyphozoan species are meroplanktonic with a bipartite life cycle.” - There is a lack of references.
Lines 322-325. “This anatomic characteristic of C. tagi distinguishes the species from the other two Rhizostomatidae, Rhizostoma luteum (Kienberger et al., 2018) and Rhizostoma octopus (Holst et al., 2007), occurring in the same geographical area.” – Please specify, what is known about velar canals of Rhizostoma luteum and Rhizostoma octopus, how exactly are they differ.
Line 363. “(Gueroun, unpublished data)” – the date is also required.
Lines 363-368. It would be great if you’ll describe the dynamics of salinity and temperature changes during the year in Tagus estuary.

3) Structure of the article is clear. There are just few notes:
Abstract lacks the conclusion.
Between the lines 244 and 245 there is a lack of introduction. Stages of what are you going to describe?
Lines 308-310. “Our observations of adult gonads from individuals collected in the Tagus estuary (Portugal) showed the absence of planulae” – This information wasn’t mentioned in “results”.
Lines 359-360. “C. tagi scyphistomae were repeatedly strobilated twice a month (≥18°C), producing up to up to 15 ephyrae (6.8 ± 3.6)” – The same.

Table 1 and Table 2 would be better in a form of complex Figures with letters on each photo. Thus you will be able to make more concrete references in the text.
Also margins of the photos in Tables 1 and 2 should be corrected: less black space is needed and the color of the background must be well matched (in the current state the slight difference between colors is seen).
Table 4 is very informative, it is a pity that the information in this table isn’t discussed in the text.
The necessity of Figure 2 is questionable.
The color coding on Figure 7 is ambiguous.

Experimental design

Some notes on Materials and methods:
Line 112. “containers of ultraviolet treated artificial seawater (salinity 35) at 18°C” – What artificial seawater was used in experiments? What brand or the exact composition if it is self-made?
It is unclear, how much food was used per feeding.
What was the approximate number of planulae, used in experiment? – it should be mentioned in the text.
Lines 172-173. “Specimens in the unfed treatment never received food.” – Was the protocol of water exchange the same for this group?

Validity of the findings

How many fully developed polyps had 18 tentacles? It doesn’t seem plausible that this number was constant, because according to the literature (for example, articles that you cite in Table 4) Scyphozoa polyps usually have number of tentacles, which is multiple of four. Some individuals have extra tentacles or lack few tentacles but most have 16 or 32 tentacles. Please check your data.
Lines 245-267 and Table 2. Please add the timing of the stages. Most likely it isn’t the exact timing, but rather time ranges, nevertheless they should be given.

Additional comments

The research is important, well established and informative (Only the number of tentacles is questionable and the timing of the stages is missing). But unfortunately, the text of the article is poorly written and definitely should be revised.

---

## Round 0.2 · accepted · Accept

There are too few people studying the biology of Cnidaria. The authors have paid careful attention to the reviewers' comments.

If we need to clarify any details required to move the manuscript forward, then our production staff will get in touch with you. Otherwise, a proof will be forthcoming shortly for your review.

Congratulations and thank you for your submission.

---

## Author Rebuttal · Round 0.2

**Sonia K.M. Gueroun**
MARE- Marine and Environmental Sciences
Centre, Quinta do Lorde Marina
Sítio da Piedade
9200-044 Caniçal, Madeira, Portugal

**Email**: sgueroun@mare-centre.pt
**Web**: www.canning-clode.com

July 24th, 2021

Prof. Alexander Ereskovsky

Editor for *PeerJ* journal

I attached the revised version of a manuscript entitled "*Life cycle of the jellyfish Catostylus tagi (Scyphozoa: Rhizostomeae) with notes on the species environmental plasticity Catostylus tagi (Class: Scyphozoa, Order: Discomedusae, Suborder: Rhizostomida, Family: Catostylidae) life cycle and first insight into its ecology"* by Sonia KM Gueroun, Tatiana M Torres, Antonina dos Santos, Nuno Vasco-Rodrigues, João Canning-Clode and Carlos Andrade for consideration of publication in the *PeerJ*. A major revision was required.

We have considered the reviewers' comments and made as suggested substantial revisions that greatly improved the quality of the manuscript. We appreciate the time and effort you and the reviewers dedicated to providing feedback on our manuscript and are grateful for the insightful comments on improvements to our paper. We have incorporated the suggestions made by the reviewers. Those changes can be followed within the manuscript using track changes. Additionally, we replied to the reviewers' comments in blue and prefaced by "Author's reply". Finally, the revised manuscript was checked by a professional proofreading service. A certificate is joined as proof in the end of this document.

The authors have read the manuscript and agreed on submission to *Peer J*.

We look forward to hearing from you regarding our submission. We would be glad to respond to any further questions and comments that you might have..

Sincerely,

Sonia K.M. Gueroun

**Sonia K.M. Gueroun**
MARE- Marine and Environmental Sciences
Centre, Quinta do Lorde Marina
Sítio da Piedade
9200-044 Caniçal, Madeira, Portugal

**Email**: sgueroun@mare-centre.pt
**Web**: www.canning-clode.com

**Reviewer 1**

*Basic reporting*

As a not native English speaking person I cannot appreciate true value of the language, but my opinion is the English can be improved.

The rest is Ok, except the figures, some of which need editting in lebels.

**The author's reply:** Thank you for your remarks. The revised manuscript was submitted to a professional proofreading service. A certificate was joined to the new submission. The labels of the figures and the titles were corrected as requested, except for figure 4: the picture was taken when the medusa was in a Kreiseil aquarium. No scaling was added in the tank to have the right size of the individuals.

*Validity of the findings*

Everything is Ok, but discussion should be improved.

**The author's reply:** Thank you for your remarks. The discussion was indeed improved

*Comments for the Author*

ABSTRACT

The last paragraph of the abstract describes the methodology and results in short, excessively short. It must present more results in more details.

**The author's reply:** the abstract was improved and more details regarding the results were added.

In the Results it is a long and detailed description of the ephyra development (growth)

**Sonia K.M. Gueroun**
MARE- Marine and Environmental Sciences
Centre, Quinta do Lorde Marina
Sítio da Piedade
9200-044 Caniçal, Madeira, Portugal

**Email**: sgueroun@mare-centre.pt
**Web**: www.canning-clode.com

stages, missing the temporal parameters. And no discussion of these data in Discussion. The authors need either discuss (compare) their data with the similar descriptions of other species, or remove such detailed descriptions.

**The author's reply:**  As the timing of the developmental stages of ephyrae is affected by environmental conditions (e.g., temperature, feeding, salinity), the stages are listed in chronological order without a designated time period.  The discussion regarding the development of the gastric stage was added Lines 342-350.

The final part - the spatial and seasonal distribution of the species (in the Results and Discussion) - in its present description sound poorly connected with the rest of the manuscript. One possibility - is to connect the revealed ecological preferences of the species with its records in the region. At the moment it sounds vice-versa.

**The author's reply:** Thank you for the remarks. We improved the discussion section regarding this subject and showed the connection between the results of the experiment, observations done during the broodstock rearing and the spatial-temporal distribution of C. tagi.

lines 1, 38, 42, 348, Table 4 - Rhizostomida -you should clearly define the taxonomic classification used in the study. According to WoRMS taxon names are different.

Moreover sometimes the spelling is different in different parts of the manuscript: Rhizostomida or Rhizostomidae...

**The author's reply:** In WORMS and the World Atlas of Jellyfish published by Jarms and Morandini  (2019), Rhizostomida is a suborder. The taxonomic level is mentioned in the revised manuscript and the spelling is homogenised.

**Sonia K.M. Gueroun**
MARE- Marine and Environmental Sciences
Centre, Quinta do Lorde Marina
Sítio da Piedade
9200-044 Caniçal, Madeira, Portugal

**Email**: sgueroun@mare-centre.pt
**Web**: www.canning-clode.com

*Other comments you will find in 'comments' in the file.*

**The author's reply:**  the various comments and remarks were taken into account. The spelling mistakes were corrected and the questions

*More detailed replies for some comments in the Annotated manuscript:*

**Line 112.**

*Reviewer:* What is the salinity and temperature of the natural seawater in this region?

**The author's reply:** The Tagus estuary presents a high spatial and temporal variability in the water mass with different temperatures and salinities that could vary from spatially for the same period (Neves 2010; Rodrigues et al. 2017). Salinity varies from 1 - 37‰ and temperature from 8 to 26°C. These variations can be observed seasonaly but also spatialy (Neves 2010). The fertilization was performed at room temperature and using the artificial water produced by the oceanarium of Lisbon.

**Line 119.**

*Reviewer:* why the temperature is higher compared to fertilisation?

**The author's reply:** Thank you for pointing out this mistake. It was due to confusion between rearing conditions in two different facilities. The correct temperature is 18C for the present case. The value was corrected in the revised manuscript.

**Line 144**

*Reviewer:* perhaps, too few for good statistics.

**The author's reply:** not statistic analysis were perfromed on the size of the polyps and the ephyrae.

**Sonia K.M. Gueroun**
MARE- Marine and Environmental Sciences
Centre, Quinta do Lorde Marina
Sítio da Piedade
9200-044 Caniçal, Madeira, Portugal

**Email**: sgueroun@mare-centre.pt
**Web**: www.canning-clode.com

**Line 144**

*Reviewer:* perhaps, too few for good statistics.

**The author's reply:** As we were shorten in polyp to perform more treatment, we restricted to two temperatures 15 - 20°C which is the usual common temperature used in the JF culture in the Oceanario but also reflects values recorded in the Tagus. We recognize that for more ecological information of the species, additional treatments are needed. That's why we mention in the title "first insight of its ecology"). Hopefully, in the future, more extended experiments also involving salinity will be performed to enrich our knowledge on this species.

**Line 235**

*Reviewer:* You just mentioned that an average of $6.8 \pm 3.6$ ephyrae per strobilation were produced

**The author's reply:** The $6.8 \pm 3.6$ ephyrae is the average calculated from all the strobilation events. The $4 \pm 1.7$ and $3.7 \pm 0.6$ ephyrae per strobilation are calculated only from the polyps that performed two successive strobilations.
* * *
**Reviewer 2**

*Basic reporting*

This research, even if it contains original data of undoubted interest, has some criticalities.

English would need to be refined. References ok.

Article structure should be improved; the morphological/morphometric description is sufficient, but the section concerning the experiments is a bit confusing and unclear. Fig. 7 is unclear and confusing

There are also some doubts about the experimental design system, see next point.

**Sonia K.M. Gueroun**
MARE- Marine and Environmental Sciences
Centre, Quinta do Lorde Marina
Sítio da Piedade
9200-044 Caniçal, Madeira, Portugal

**Email**: sgueroun@mare-centre.pt
**Web**: www.canning-clode.com

**The author's reply:** Thank you for your remarks. The revised manuscript was submitted to a professional proofreading service. A certificate was joined to the new submission. The section of the experiment results were clarified. Figure 7 was split into a figure and a table 2 for more clarity.

*Experimental design*

The authors tested the effect of temperature and salinity on planula development, and this is OK, but why didn't they perform the same type of experiments on the asexual reproductive rate of the polyps?

**The author's reply:** Thank you for pointing out this. An experiment on the effect of different salinity-temperature regimes was planned. However, not enough polyps were available to cover the same treatment groups as the planulae. So, to cover the same range, we postponed the experiment until we get enough polyps. However, due to the COVID-19 situation, the rearing was shut down and the broodstock was lost. We were, therefore, unable to perform this trial. We hope to be able to achieve this experiment in the future. Despite this, we strongly believe that the present manuscript present important pionee information regarding *C. tagi* ecology.

The authors limited themselves to analyzing the effect of temperature and feeding on asexual reproduction, but this section is somewhat confusing and unclear.

As for the statistical analysis, some doubts fall in the methodological approach adopted. Following the authors' rationale, the nature of repated measurements (line 194) have not been considered in the models. GLMM or LMM would be appropriate in such type of data.

**The author's reply:** For the planulae experiment, the results, as mentioned in the manuscript, were binary output; 0: planula not settled; 1: planula settled. We analysed the total time spent by each planula in the planktonic stage. We used the similar statistical approach used by Takao and Uye (2018) for the same experiment. To make our statistic choice clearer, we added the reference in the M&M section.

**Sonia K.M. Gueroun**
MARE- Marine and Environmental Sciences
Centre, Quinta do Lorde Marina
Sítio da Piedade
9200-044 Caniçal, Madeira, Portugal

**Email**: sgueroun@mare-centre.pt
**Web**: www.canning-clode.com

Furthermore, in relation to the performed models, I think that, once a Poisson GLM model was overdispersed, the Authors have to opt for a quasipoisson distribution of error gust changing the overdispersion parameters) since prediceted values (integers) are more realistic than those predicted by a negative binomial distribution.

In general, the whole section describing statistical methods can be improved and Authors choices better justified.

**The author's reply:** The first overdispersions detected in the GLMs models were clearly driven by the excess of zero (72.2% of the data were zero). As advised by the reviewer, we performed a zero-inflated model with a quasi-Poisson family using glmmTMB r package. Then compared the AIC and BIC for ZIP, ZINB and ZiqP model to select the best model.
* * *
**Reviewer 3**

*Basic reporting*

1)      I would strongly recommend this article to be revied by a native speaker. English, which is used here, is ambiguous. Abstract is especially hard to understand due to the fact that sentences are not connected with each other. Moreover, there are multiple typos and some mistakes in terminology.

Problems with language, that I can define:

Lines 1.38.42. "Rhizostomida", the preferred name of this suborder is Dactyliophorae.

Line 1. Also, it is strange to name class and suborder (Scyphozoa, Rhizostomida) without naming the order (Scyphozoa, Rhizostomeae, Dactyliophorae). Maybe naming only class and order will be more appropriate (Scyphozoa, Rhizostomeae).

Lines 32-33 are unreadable. What do you mean by "jellyfish outbreaks are ... natural events"? What do you mean by "environmental... activities"? How the first leads to the second?

**Sonia K.M. Gueroun**
MARE- Marine and Environmental Sciences
Centre, Quinta do Lorde Marina
Sítio da Piedade
9200-044 Caniçal, Madeira, Portugal

**Email**: sgueroun@mare-centre.pt
**Web**: www.canning-clode.com

Line 34. "different jellyfish life stage" — it should be: "different jellyfish life stages" Line 34.385. "its" — "it's"

Line 35. "have its full life cycle understood" — I'm not sure the word "understood" is a good fit for the scientific article.

Line 38. "scyphistoma presents 18 tentacles" — The word "presents" doesn't look like a good choice.

Line 39. "Only podocyst production was observed." — This sentence should be expanded. In the current state it conflicts with the next sentence, where you talk about strobilation, that you also observed.

Line 41. "ephyrae... at liberation" — The word "detachment" or "release" is usually used.

Lines 52-53. "Tagus estuary" is used twice.

Lines 56-57. What do you mean by "The outbreaks often have severe... ecosystems"? Please rephrase.

Lines 57-60. The sentence is too complicated and thus very hard to understand. Lines 63.115.157. I would recommend to use the word "attached" instead of "fixed". Lines 64-65.224-225.326-327.332-333. "(e.g. budding, swimming buds, podocysts, fission, strobilation) (Arai, 1997)"; "lateral budding by stolon, lateral scyphistoma budding, or pedalocysts"; "various budding modes (e.g. lateral budding, budding from stolon, motile bud-like tissue particles, etc.) and podocysts (Arai, 1997)"; "lateral budding, lateral budding through stolons, reproduction from parts of stolons/stalks, motile bud-like tissue particles, podocysts) (Schiariti et al., 2014)" — The unification of the names of modes of asexual reproduction is required, especially while you cite the same article on this lines.

Line 68. "both phases integral to the study" — I'm not sure the word "integral" fits here. Lines 112.119.148.149.150.154.271.273.277.280.282.350.366.377.Figure 5.Gigure 6.

Please use the sign %o , when you specify the salinity. Line 115. "given to planula choice to" — please rephrase.

[Figure]

**Sonia K.M. Gueroun**
MARE- Marine and Environmental Sciences
Centre, Quinta do Lorde Marina
Sítio da Piedade
9200-044 Caniçal, Madeira, Portugal

**Email**: sgueroun@mare-centre.pt
**Web**: www.canning-clode.com

Lines 140.230.237-238.249.320.Table 1.Figure 1.Figure 2. "ephyra lappet elongated";

"Ephyrae typically had eight marginal arms with a pair of antler palm-like rhopalial lappets and a single rhopalium per lappet" — unification of the terms is needed. Which part of ephyra do you call "marginal arm", "marginal lobe", "lappet", "marginal lappet", "rhopalial lappet", "rhopalia lappet"?

Line 142. "MDD/CL 9 100" — I believe it was supposed to be "MDD/CL x 100". Lines 143.144. "CDD/TBD 9 100", "RLL/TMLL 9 100" — the same.

Line 148. "three temperature" — "three temperatures".

Line 161. "The effect of temperature and feeding on asexual reproduction" — "The effect of temperature and feeding regime on asexual reproduction".

Line 162. "two temperature levels" — "two temperatures". Line 163. "three feed regimes" — "three feeding regimes".

Lines 163-164. "Eighteen polyps were tested in each of the six combinations." — This sentence is ambiguous and thus should be rephrased.

Lines 164-165. The sentence is very hard to understand. Please rephrase.

Line 215-216. 218. "total body length (TBD)" — I believe it was supposed to be "total body length (TBL)".

Line 218-219. "was distinctly longer than wide" — seems to be a poor choice of words, specifics are required.

Line 227. "strobiles" — "strobilae" is preferable.

Lines 229-230. "calyx upper part" — How do you define "up" and "down" in strobila? It would be correct to use terms "oral"-"aboraI".

Lines 245-248. "The first oral tentacles develop on the distal ends of the manubrium.

[Figure]

**Sonia K.M. Gueroun**
MARE- Marine and Environmental Sciences
Centre, Quinta do Lorde Marina
Sítio da Piedade
9200-044 Caniçal, Madeira, Portugal

**Email**: sgueroun@mare-centre.pt
**Web**: www.canning-clode.com

... The first oral tentacles develop around the manubrium opening" — information seems to be doubled.

Line 258.Table 2. "two other canals develop horizontally" — the word "horizontally" doesn't describe the position of the structure in the jellyfish body. Please rephrase. Line 260. The word "canals" is missing.

Line 359. "up to" is used twice. Table 1. "ep:ephyrae" — "ep:ephyra". "ephyrae:500pm" - "ephyra:500pm" "ephyrae" - "ephyra"

"•18 filiform tentacles •Long, club-shape and flexible" — I believe it was supposed to be "•18 filiform tentacles; long, club-shape and flexible"

"•MDD = 44 % of the TBL" — Is everything correct here? Diameter in relation to length?

**The author's reply:** Thank you for pointing out these mistakes. The calculation was corrected.

Table 2. "Development stages" - "Developmental stages".

"• Rhopalia canals are sli forked, rounded points • • rhopalial canals slightly forked with rounded points" — information is doubled.

Figure 3. "Enlargement of the mouth development of Catostylus tagi." — "Enlargement

of the mouth of Catostylus tagi."

"at the lips mouth in (stage 1)" — the content is not clear.

Figure 5. "Stage 1: 1-3 tentacles" — but on the lines 157-158 "Stage 1: 1-4 tentacles". Figure 7. "C. tagi asexual reproduction under temperature and diet regimes." — the word "different" seems to be missing.

"ephyra. strobiIation-1" - "ephyrae.strobiIation-1".

**The author's reply:** Thank you for pointing out all the mistakes. The corrections were performed, the terms unified, and the complex/confusing sentences rephrased. Moreover, the revised

**Sonia K.M. Gueroun**
MARE- Marine and Environmental Sciences
Centre, Quinta do Lorde Marina
Sítio da Piedade
9200-044 Caniçal, Madeira, Portugal

**Email**: sgueroun@mare-centre.pt
**Web**: www.canning-clode.com

manuscript was submitted to a professional proofreading service. A certificate was joined to the new submission.

2)     The background is present, but it lacks the known information on ephyra development. This work is not the first to show the development of gastric system and margin of the Scyphozoan young medusa. For example:

•       Holst, S. (2012). Morphology and development of benthic and pelagic life stages of North Sea jellyfish (Scyphozoa, Cnidaria) with special emphasis on the identification of ephyra stages. Marine Biology, 159(12), 2707—2722. doi:10.1007/s00227-012- 2028-0

•       Holst, S., Sötje, I., Tiemann, H., & Jarms, G. (2007). Life cycle of the rhizostome jellyfish Rhizostoma octopus (L.) (Scyphozoa, Rhizostomeae), with studies on cnidocysts and statoliths. Marine Biology, 151(5), 1695—1710. doi:10.1007/s00227- 006-0594-8 (Which is by the way cited in the text).

Lines 61-62. "With some exceptions (e.g. Pelagia noctiluca), Scyphozoan species are meroplanktonic with a bipartite life cycle." - There is a lack of references.

Lines 322-325. "This anatomic characteristic of C. tagi distinguishes the species from the other two Rhizostomatidae, Rhizostoma luteum (Kienberger et al., 2018) and Rhizostoma octopus (Holst et al., 2007), occurring in the same geographical area." — Please specify, what is known about velar canals of Rhizostoma luteum and Rhizostoma octopus, how exactly are they differ.

Line 363. "(Gueroun, unpublished data)" — the date is also required.

Lines 363-368. lt would be great if you'll describe the dynamics of salinity and temperature changes during the year in Tagus estuary.

**The author's reply:** Thank you for your remarks. The section regarding the anatomic comparison was improved. For Line 363: as mentioned, this is unpublished data from personal observation from the culture. Therefore, no data will be added.

3)       Structure of the article is clear. There are just few notes:

**Sonia K.M. Gueroun**
MARE- Marine and Environmental Sciences
Centre, Quinta do Lorde Marina
Sítio da Piedade
9200-044 Caniçal, Madeira, Portugal

**Email**: sgueroun@mare-centre.pt
**Web**: www.canning-clode.com

*Abstract lacks the conclusion.*

Between the lines 244 and 245 there is a lack of introduction. Stages of what are you going to describe?

Lines 308-310. "Our observations of adult gonads from individuals collected in the Tagus estuary (Portugal) showed the absence of planulae" — This information wasn't mentioned in "results".

Lines 359-360. "C. tagi scyphistomae were repeatedly strobilated twice a month (a18°C), producing up to up to 15 ephyrae (6.8 + 3.6)" — The same.

**The author's reply:** Thank you for your remarks. Corrections were made. Regarding the line308-310, as mentioned in the M&M, the gonads of the different specimens were observed to identify the male, the female and the level of maturation. Naturally, if planulae were present in the gonads, they would be noted.

Table 1 and Table 2 would be better in a form of complex Figures with letters on each photo. Thus you will be able to make more concrete references in the text.

Also margins of the photos in Tables 1 and 2 should be corrected: less black space is needed and the color of the background must be well matched (in the current state the slight difference between colors is seen).

Table 4 is very informative, it is a pity that the information in this table isn't discussed in the text.

The necessity of Figure 2 is questionable. The color coding on Figure 7 is ambiguous.

**The author's reply:** Thank you for your remarks. The table 1 and 2 were merged in one complex figure (Fig 2) and each picture was mentioned in the text; the picture baground was homogenised. The figure 7 was modified, splited into one figure and one table (Table 2). The discussion regarding the gastric system and the ephyra anatomy to distinguish between different species (occurring in the same area) was improved.

*Experimental design*

**Sonia K.M. Gueroun**
MARE- Marine and Environmental Sciences
Centre, Quinta do Lorde Marina
Sítio da Piedade
9200-044 Caniçal, Madeira, Portugal

**Email**: sgueroun@mare-centre.pt
**Web**: www.canning-clode.com

Some notes on Materials and methods:

Line 112. "containers of ultraviolet treated artificial seawater (salinity 35) at 18°C" — What artificial seawater was used in experiments? What brand or the exact composition if it is self-made?

**The author's reply:** The brand was added in the revised manuscript

It is unclear, how much food was used per feeding.

**The author's reply:** As the trial was on the type of food and not on the quantity, the polyps were fed in excess for a fixed period of time (1.5h).

What was the approximate number of planulae, used in experiment? — it should be mentioned in the text.

**The author's reply**: Thank you for pointing it out. In the revised manuscript, the number of planuae are clearly mentioned in line 157 (while in the first manuscript, this number was indirectly mentioned by the mumber of 6-well plate used).

Lines 172-173. "Specimens in the unfed treatment never received food." — Was the protocol of water exchange the same for this group?

**The author's reply:** Thank you for the remark. The unfed treatment group followed the same water change as the other groups. This is now mentioned in the revised manuscript.

*Validity of the findings*

How many fully developed polyps had 18 tentacles? It doesn't seem plausible that this number was constant, because according to the literature (for example, articles that you cite in Table 4) Scyphozoa polyps usually have number of tentacles, which is multiple of four. Some individuals have extra tentacles or lack few tentacles but most have 16 or 32 tentacles. Please check your data.

**Sonia K.M. Gueroun**
MARE- Marine and Environmental Sciences
Centre, Quinta do Lorde Marina
Sítio da Piedade
9200-044 Caniçal, Madeira, Portugal

**Email**: sgueroun@mare-centre.pt
**Web**: www.canning-clode.com

**The author's reply:** Thank you for pointing out this careless mistake during the writing process. In fact, the polyps have 16 tentacles.

Lines 245-267 and Table 2. Please add the timing of the stages. Most likely it isn't the exact timing, but rather time ranges, nevertheless they should be given.

**The author's reply:** As the timing of the developmental stages of ephyrae are affected by environmental conditions (e.g., temperature, feeding), the different stages are given without desgined time.

*Comments for the Author*

The research is important, well established and informative (Only the number of tentacles is questionable and the timing of the stages is missing). But unfortunately, the text of the article is poorly written and definitely should be revised.

**The author's reply:** the manuscript was revised, the term homogenized, and the final revised manuscript was submitted to a professional proofreading service for spelling mistakes and comprehension of the text. A certificate is joined to the present manuscript.

[Figure]

Proof-Reading-Service.com Ltd, Devonshire
Business Centre, Works Road, Letchworth Garden
City, Hertfordshire, SG6 1GJ, United Kingdom
Office phone: +44(0)20 31 500 431
E-mail: enquiries@proof-reading-service.com
Internet: http://www.proof-reading-service.com
VAT registration number: 911 4788 21
Company registration number: 8391405

21 July 2021

To whom it may concern,

**RE: Proof-Reading-Service.com Editorial Certification**

This is to confirm that the document described below has been submitted to Proof-Reading-Service.com for editing and proofreading.

We certify that the editor has corrected the document, ensured consistency of the spelling, grammar and punctuation, and checked the format of the sub-headings, bibliographical references, tables, figures etc. The editor has further checked that the document is formatted according to the style guide supplied by the author. If no style guide was supplied, the editor has corrected the references in accordance with the style that appeared to be prevalent in the document and imposed internal consistency, at least, on the format.

It is up to the author to accept, reject or respond to any changes, corrections, suggestions and recommendations made by the editor. This often involves the need to add or complete bibliographical references and respond to any comments made by the editor, in particular regarding clarification of the text or the need for further information or explanation.

We are one of the largest proofreading and editing services worldwide for research documents, covering all academic areas including Engineering, Medicine, Physical and Biological Sciences, Social Sciences, Economics, Law, Management and the Humanities. All our editors are native English speakers and educated at least to Master's degree level (many hold a PhD) with extensive university and scientific editorial experience.

**Document title: Catostylus tagi (Class: Scyphozoa, Order: Discomedusae, Suborder: Rhizostomida, Family: Catostylidae) life cycle and first insight into its ecology**

**Author(s):**  **Sonia KM Gueroun, Tatiana M Torres, Antonina dos Santos, Nuno Vasco-Rodrigues, João Canning-Clode, Carlos Andrade**

**Format:**  **British English**

**Style guide:**  **Style guide has been supplied at https://peerj.com/about/author-instructions/**